# Exploration of the Effect on Genome-Wide DNA Methylation by *miR-143* Knock-Out in *Mice* Liver

**DOI:** 10.3390/ijms222313075

**Published:** 2021-12-03

**Authors:** Xingping Chen, Junyi Luo, Jie Liu, Ting Chen, Jiajie Sun, Yongliang Zhang, Qianyun Xi

**Affiliations:** Guangdong Provincial Key Laboratory of Animal Nutrition Control, National Engineering Research Center for Breeding Swine Industry, College of Animal Science, South China Agricultural University, No. 483 Wushan Road, Guangzhou 510642, China; cxp0315@stu.scau.edu.cn (X.C.); luojunyi@scau.edu.cn (J.L.); 20181069001@stu.scau.edu.cn (J.L.); allinchen@scau.edu.cn (T.C.); jiajiesun@scau.edu.cn (J.S.)

**Keywords:** DNA methylation, *miR-143*, Whole-Genome Bisulfite Sequencing, liver diseases

## Abstract

*MiR-143* play an important role in hepatocellular carcinoma and liver fibrosis via inhibiting hepatoma cell proliferation. *DNA methyltransferase 3 alpha* (*DNMT3a*), as a target of *miR-143*, regulates the development of primary organic solid tumors through DNA methylation mechanisms. However, the effect of *miR-143* on DNA methylation profiles in liver is unclear. In this study, we used Whole-Genome Bisulfite Sequencing (WGBS) to detect the differentially methylated regions (DMRs), and investigated DMR-related genes and their enriched pathways by *miR-143*. We found that methylated cytosines increased 0.19% in the *miR-143* knock-out (KO) liver fed with high-fat diet (HFD), compared with the wild type (WT). Furthermore, compared with the WT group, the CG methylation patterns of the KO group showed lower CG methylation levels in CG islands (CGIs), promoters and hypermethylation in CGI shores, 5′UTRs, exons, introns, 3′UTRs, and repeat regions. A total of 984 DMRs were identified between the WT and KO groups consisting of 559 hypermethylation and 425 hypomethylation DMRs. Furthermore, DMR-related genes were enriched in metabolism pathways such as carbon metabolism (*serine hydroxymethyltransferase 2* (*Shmt2*)*, acyl-Coenzyme A dehydrogenase medium chain* (*Acadm)*), arginine and proline metabolism (*spermine synthase* (*Sms*), *proline dehydrogenase* (*Prodh2*)) and purine metabolism (*phosphoribosyl pyrophosphate synthetase 2* (*Prps2)*). In summary, we are the first to report the change in whole-genome methylation levels by *miR-143*-null through WGBS in *mice* liver, and provide an experimental basis for clinical diagnosis and treatment in liver diseases, indicating that *miR-143* may be a potential therapeutic target and biomarker for liver damage-associated diseases and hepatocellular carcinoma.

## 1. Introduction

*MiR-143*-encoding genes are highly conserved and located on the fifth autosome [1]. MicroRNAs (miRNAs) are endogenous short single-stranded non-coding RNA molecules, found in eukaryotic cells, that range in lengths from 18 to 25 nt [2,3]. MiRNAs have been found to play a crucial role in the regulation of all major cellular functions, including cell proliferation, differentiation, and apoptosis, which are involved in various *human* diseases [4]. *MiR-143* is widely distributed in mammalian tissues and serves an important role in a number of physiological processes, such as adipocyte and smooth muscle differentiation [5,6,7], tumorigenesis suppression [8], DNA methylation [9,10,11,12] and development of tissues and organs [13]. *MiR-143* has been reported as a biomarker for the late stages of hepatocellular carcinoma and liver fibrosis [14,15]. MiRNA degrades the target mRNA, or inhibits its translation, by combining with the seed region in the 3′ UTR of the target mRNA, hence being able to regulate physiological or biochemical processes [16]. *MiR-143* enhances hepatocarcinoma metastasis by repressing fibronectin expression [17]. *MiR-143* plays a role as tumor suppressor via inhibiting hepatoma cell proliferation [18,19,20,21]. *MiR-143* induces apoptosis of liver carcinoma cells through the regulation of the *NF-κB* pathway [22]. Bone marrow macrophage-derived exosomal *miR-143-5p* induces insulin resistance in hepatocytes through repressing *mitogen-activated protein kinase phosphatase-5* (*MKP5*) [23]. Overexpression of *miR-143* inhibits insulin-stimulated *AKT* activation and impairs glucose metabolism by targeting *oxysterol-binding protein related protein* (*ORP8*) in the liver of obese *mice* models [24]. However, these data have not been investigated for DNA methylation in those studies. 

Many studies have shown that miRNAs are involved in epigenetic modifications [25,26,27,28,29]. Epigenetic modifications such as methylation/demethylation have been shown to be involved in metabolism and diseases of the liver [30]. The definition of epigenetic modification is the heritable change in gene expression or function without alterations in the DNA sequence [31,32]. One of the most widely studied epigenetic modifications is DNA methylation [33]. DNA methylation plays an important role in regulating chromatin structure and therefore regulating gene expression. Generally, DNA methylation is a process that transfers a methyl group to the 5′ position of cytosine to form 5-methylcytosine (5mC), catalyzed by *DNA cytosine-5-methyltransferases* (*DNMTs*) [34]. In mammalian cells, DNA methylation occurs on the 5′ position of cytosine and almost exclusively on cytosine-guanine dinucleotides (CpG) [35]. DNA methylation usually suppresses gene expression, whereas DNA demethylation can lead to the reactivation and expression of the gene or the activation of translocation [36]. Various physiological or biochemical processes are regulated by specific DNA methylation patterns including transcription silencing, transposon repression, cell differentiation, genomic imprinting, the inactivation of the X chromosome and the regulation of tissue-specific gene expression [37]. Therefore, DNA methylation plays a critical role in the regulation of gene expression, genomic DNA stability, cell proliferation, and malignant transformation [38]. The alteration of DNA methylation modification in chronic liver disease progression is important and helpful to find biomarkers of liver disease in clinical diagnosis and therapies [39]. 

Previous researches have shown that *DNMT3a*, a DNA methyltransferase contributing to de novo methylation, is a target of *miR-143* [9,10,11,12]. These studies have reported on solid tumors of primary organs other than the liver. Recently, some studies have investigated the epigenetic alterations induced by chemicals and toxicants which mediate the alteration of the expression levels of miRNAs and the DNA methylation trying to integrate these different epigenetic modifications and associate them with the development of diseases [40,41,42]. However, the effect of *miR-143* on whole-genome methylation is unclear in *mice* liver, such as the change in methylation level in the liver genome, the differentially methylated regions (DMRs) and DMR-related genes and their enriched pathways.

Bioinformatics is an interdisciplinary subject that includes the life sciences, mathematics, and computer science [43]. Its research methods are used for mining and understanding the biological significance contained in massive data through various technologies and tools of computer science, biology and mathematics [44]. Many studies have shown that bioinformatics-based DNA methylation analysis is widely used for the identification of multiple human diseases, malignant tumor diagnosis, biomarker screening and targeted therapy [45,46,47,48,49,50]. The gold-standard method for quantitative interrogation of the methylation state of all CpG dinucleotides in a genome is Whole-Genome Bisulfite Sequencing (WGBS) [51]. WGBS is an unbiased method for genome-wide DNA methylation profiling [52]. Clustered regularly interspaced short palindromic repeats (CRISPR)–*CRISPR-associated protein 9* (*Cas9*)-mediated genome modification is a rapid and efficient tool for editing the genomes of a variety of organisms [53,54]. This technique is widely used to explore developmental mechanisms, gene function and expression regulation, etc. [55,56].

In this study, we used CRISPR/Cas9 to generate *miR-143* knock-out (KO) transgenic *mice* to explore the roles of *miR-143* on DNA methylation by Whole-Genome Bisulfite Sequencing. Our goal is to dissect the differentially methylated regions and decipher the correlation of *miR-143* and DNA methylation and how they regulate genomic gene expression and explore the role in liver disease to provide an experimental basis for clinical diagnosis and treatment.

## 2. Results

### 2.1. Identification of Transgenic Mice

As shown in Figure 1, the electrophoretic bands and gene sequence alignment show that an ~105 bp fragment containing *miR-143*-encoding gene was deleted in *miR-143* KO *mice*. *MiR-143* was not expressed in *miR-143 KO mice* liver (Figure 1D). The above results showed that *miR-143* transgenic *mice* were established successfully.

### 2.2. WGBS Roundup

A total of 242.21 Gb data and 807,355,041 reads were obtained by Whole-Genome Bisulfite Sequencing (WGBS). Strict quality control was conducted for each sample to evaluate whether the sequencing data are qualified. After quality control, 789,789,445 clean reads (215.31 Gb) were obtained, and 68.13% (WT), 72.41% (KO) of the clean reads were uniquely mapped to the *mouse* reference genome (*Mus musculus*, *GRCm38.p6*; https://www.ncbi.nlm.nih.gov/genome/?term=Mus+musculus, accessed date: 6 April 2019). In the methylation assay, one of the important indicators to evaluate the sequencing depth is the level of cytosine (C) sites coverage. The average C site was 10,010.9 (WT), 11,945.9 (KO) Mb and 383.3 (WT), 462.4 (KO) Mb CG sites (Table 1). The coverage rate for each chromosome and the coverage of C sites in each context (CG, CHH, and CHG) were uniform in WT and KO sample (Figure 2A–F). The DNA methylation density and DNA methylation level are shown in Figure 2G,H. We calculated the methylation level of each C site, and we found that there were 1,143,203,528 C sites in the *mouse* genome, 32,321,811 (2.82%) methylated in the WT group, and 34,460,213 (3.01%) methylated in the KO group. Under a HFD diet, the methylated C sites consisted of 93.73% mCG, 1.04% mCHG, and 5.22% mCHH in the WT group (Figure 2I). However, after *miR-143* KO, the methylated C sites included 92.29% mCG, 1.29% mCHG, and 6.41% mCHH (Figure 2J). Compared with the WT group, methylated cytosines increased by 0.19% in the KO group.

Motif analysis is important in the determination of DNA-protein binding sites. For each sequence context (CG, CHG, and CHH), we analyzed its genome-wide environmental site information and the methylated site sequence information (9 bp including sites) to determine the enrichment of particular local sequences [57]. We found a prominent CAG and a CAC sequence motif at the CHG and CHH methylated sites, respectively (Figure 3). This result is consistent with previous findings reported in mammals [58,59].

The distribution of 5-methylcytosine level in sequence contexts, gene density and DNA methylation level across chromosomes in the liver of the WT and KO groups were shown in Figure 4A–D. Compared with the WT group, the CG methylation patterns of the KO groups showed lower CG methylation levels in CG islands (CGI) and promoters, and hypermethylation in CGI shores, 5′UTRs, exons, introns, 3′UTRs, and repeat regions (Figure 4E). CG methylation density in regions except CGI were lower in the KO group than in the WT group (Figure 4F). In the contexts of both CHG and CHH, the patterns of methylation level and density in the CGIs, CGI shores, promoters, 5′UTRs, exons, introns, 3′UTRs and repeat regions were largely different between KO and WT group. The heat map showed the methylation levels in the gene functional regions in the WT and KO groups (Figure 4G–I).

### 2.3. Characterization of DMRs

A total of 984 DMRs were compared in the WT and KO groups, in which 559 hypermethylation and 425 hypomethylation DMRs were found in the liver of KO *mice* compared with WT *mice*. A heat map was generated by a cluster analysis of DMRs between the WT and KO groups (Figure 5A). The length distribution of DMRs ranged from dozens to hundreds of nucleotides and complied with the Gaussian distribution (Figure 5B), which is consistent with previous studies [60,61]. Violin boxplot was used to plot mean methylation levels, which showed that DMR methylation levels were mostly at medium level and not at low or high levels, and the mean methylation level was higher after *miR-143* knock-out (Figure 5C). The distribution of the DMR-anchored region was shown in Figure 5D. The results showed that DMRs were mainly distributed in the CGIs, exons and introns. A Circos plot was drawn to show the distribution and statistical significance of DMRs in each chromosome (Figure 5E).

### 2.4. Functional Enrichment Analysis: Gene Ontology (GO)

Genes that overlap with DMRs for at least 1 bp in the functional region are known as the DMR-related genes. According to DMR genome position, DMR-related genes were annotated in Appendix A. Results showed that hypermethylated DMRs were associated with 475 genes and hypomethylated DMRs were associated with 353 genes. Gene Ontology (GO, http://www.geneontology.org/, accessed date: 7 April 2019) is an international standardization of gene function classification system [62]. Based on the Wallenius non-central hypergeometric distribution, GO enrichment analysis of the DMR-related genes was implemented by the GOseq [63]. Results of GO enrichment analysis of the DMR-related genes was showed in Appendix A. As shown in Figure 5F, most of DMR-related genes were enriched significantly in the single-organism biologic process of the GO (GO:0044699, *p* = 3.05 × 10^−6^). Moreover, a lot of DMR-related genes were involved in cellular metabolic biologic process (GO:0044237, *p* = 8.71 × 10^−6^). We then noticed that many DMR-related genes were also involved in cellular developmental process (GO:0048869, *p* = 5.77 × 10^−9^) and cell differentiation (GO:0030154, *p* = 1.11 × 10^−9^). For molecular function analysis, a great many of DMR-related genes were enriched in binding (GO:0005488, *p* = 3.04 × 10^−10^) and protein binding (GO:0005515, *p* = 9.93 × 10^−7^).

### 2.5. KEGG Pathway Enrichment Analysis 

In the organism, different genes coordinate with each other to exercise their biological functions [64]. Through the significant enrichment of specific metabolic pathways, we can determine the biochemical metabolic pathways and signal transduction pathways that may be in association with DMR-related genes. The Kyoto Encyclopedia of Genes and Genomes (KEGG) is the major public pathway-related database [65]. Hypergeometric test was used to find significantly enriched KEGG pathways associated with DMR-related genes (Appendix A). As shown in Figure 5G, we found that DMR-related genes were enriched in metabolism pathways such as purine metabolism (3 hypermethylation and 9 hypomethylation genes, including *phosphoribosyl pyrophosphate synthetase 2* (*Prps2*)), carbon metabolism (5 genes is hypermethylation and 3 hypomethylation, including *serine hydroxymethyltransferase 2* (*Shmt2*), and *acyl-Coenzyme A dehydrogenase medium chain* (*Acadm*)) and arginine and proline metabolism (5 hypermethylation and 1 hypomethylation genes, including *spermine synthase* (*Sms*), and *proline dehydrogenase 2* (*Prodh2*)). We also noticed that apoptosis-related pathways (2 hypermethylation and 4 hypomethylation genes) including *B cell leukemia/lymphoma 2* (*Bcl2*) and *interleukin-1 receptor-associated kinase 1* (*Irak1*) and infectious disease were involved.

### 2.6. The Expression of DMR-Related Genes at mRNA Level

To verify the WGBS results, four DMR-related genes were randomly selected from the metabolism pathways and apoptosis pathway. As shown in Appendix A, *Irak1* and *Bcl2* are hypomethylated DMR-related genes while *Prps2* and *Shmt2* are hypermethylated DMR-related genes. The results of qPCR shown that *Prps2* (Figure 6A) and *Shmt2* (Figure 6B) were significantly downregulated in the liver of KO *mice*, compared with WT *mice*. Compared with the WT group, *Bcl2* (Figure 6D) was upregulated in the KO group (*p* = 0.173). The qPCR results correspond with WGBS.

## 3. Discussion and Conclusions

It is well known that miRNAs degrades mRNA or inhibits its translation by binding to the 3′-UTR regions of the mRNA transcript. Recently, some studies have investigated the epigenetic alterations induced by chemicals and toxicants which mediate the alteration of the expression levels of miRNAs and the DNA methylation trying to integrate these different epigenetic modifications and associate them with the development of diseases [40,41,42]. We are the first to report that the whole-genome methylation level could be influenced by *miR-143* loss in *mice* liver via WGBS. In the methylation assay, one of the important indicators to evaluate the sequencing depth is the level of coverage of cytosine (C) sites. In our study, the average C site was 10,010.9 (WT), 11,945.9 (KO)Mb and 383.3 (WT), 462.4 (KO)Mb CG sites (Table 1). In mammalian cells, DNA methylation occurs on the 5′ position of cytosine and almost exclusively on cytosine-guanidine dinucleotides (CpG) [35]. In plant cells, DNA methylation also occurs on CHH and CHG (H=A, C, T) besides CpG [66]. The coverage rate for each chromosome and the coverage of C sites in each context (CG, CHH, and CHG) were uniform in WT and KO sample (Figure 2A–F). These data showed that the sequencing quality conformed in each sample. We found that methylated cytosines increased by 0.19% after *miR-143* knock out. Lai et al. showed that the level of global DNA methylation is significantly lower in non-alcoholic fatty liver disease (NAFLD) patients than in non-NAFLD overweight participants [30]. Furthermore, the level of global DNA methylation in the liver tends to decrease with the increase in hepatic inflammation and fibrosis grade and disease progression [67]. 

Motif represents the base distribution characteristics of a 9 bp sequence upstream and downstream including mC sites. It can be a conserved sequence and may play a key role in the regulation of gene expression. Motif analysis is important in the discovery of DNA-protein binding sites. We found a prominent CAG and CAC sequence motif at the CHG and CHH methylated sites, respectively (Figure 3). This result is consistent with a previous findings reported in mammals [58,59].

In mammals, the liver is the center of glucose and lipid metabolism [68]. We found that DMR-related genes were enriched in lipid metabolism. Normal lipid metabolism is particularly important for normal liver function. Abnormal liver functions such as hepatic steatosis causes NAFLD, which leads extensive inflammation, hepatocyte apoptosis and liver damage, subsequently leading to progressive fibrosis and cirrhosis [69]. We also noticed that DMR-related genes were enriched in apoptosis pathways. In healthy conditions, apoptosis play a critical role in maintain equilibrium between cell loss and replacement [70]. However, abnormal apoptosis such as cirrhosis is the key mechanism of many diseases. *Irak1* is the one of the DMR-related genes enriched in apoptosis*. Irak1* can regulate apoptosis by *PI3K/Akt* pathway [71]. On the other hand, *miR-143* can regulate apoptosis via targeting *Bcl2* [72,73,74]. Therefore, *miR-143* might be involved in the regulation of apoptosis by directly targeting apoptosis-related genes and changing the methylation level of functional genes.

We found that DMR-related genes were enriched in many pathways of amino acid metabolism. The metabolism of amino acids includes two aspects. On the one hand, amino acids are mainly used to synthesize proteins, polypeptides and other nitrogen-containing substances. On the other hand, it can be decomposed into alpha-ketoacids, amines and carbon dioxide through deamination, transamination, combined deamination or decarboxylation. The metabolites of amino acid can supply the cell with energy and furnishes carbon skeletons for biosynthesis. *Shmt2* is the key enzyme in glycine synthesis flues by serine [75]. Many studies have shown that *Shmt2* play a key role in tumor development and prognosis [76,77,78]. Rapidly growing tumors cells have large energy demands that could be provided by metabolites of amino acids. Furthermore, many studies suggest that *miR-143* is associated with the occurrence and development of many types of cancer [79,80,81,82]. Therefore, *miR-143* might be involved in the regulation of occurrence and development of many types of cancer by the changing of amino acid metabolism, and may be a biomarker in some cancer diagnosis [14,15]. Many studies have demonstrated that *RAS* genes are important targets of *miR-143* [83,84,85,86,87,88,89,90,91,92,93]. *RAS* are well known as proto-oncogenes, which code three distinct genes (*KRAS*, *NRAS* and *HRAS*) and four distinct proteins (*KRAS4A*, *KRAS4B*, *NRAS* and *HRAS*) [94]. Since *RAS* genes play a key role in multiple tumor pathogeneses, they are potential therapeutic targets for multiple tumors. So, *miR-143* may provide a new therapeutic approach in multiple tumors.

KEGG pathway enrichment analysis of DMR-related genes also showed that purine metabolism was altered by the change in methylation level of related functional genes. *Prps* is one of the key enzymes in purine metabolism. The synthesis of phosphoribosyl-pyrophosphate, which is a substrate for purine and pyrimidine nucleoside and nucleotide synthesis, is catalyzed by *Prps*. Purine is an important base compound of nucleic acid, and the end product of its metabolism in vivo is uric acid. The abnormal metabolism of purine nucleotides is the basis of some diseases and the target of their treatment. Uric acid has been found being a prooxidant, and contributes to tumorigenesis via reactive oxygen species and inflammatory stress [95,96]. Previous studies have shown that uric acid induces hepatocyte fat accumulation [97], development of non-alcoholic fatty liver disease (NAFLD) [97,98,99,100,101], liver cirrhosis [102,103] and hepatocellular carcinoma [96,104]. In this study, we found that *miR-143* loss inhibited purine synthesis in the liver. Therefore, *miR-143* may provide a therapeutic target for the treatment of liver damage-associated diseases and hepatocellular carcinoma, but more detailed studies, such as signaling pathways, epigenetic modifications, and enzyme regulation, need to be explored further. 

With their unique properties, including low immunogenicity, innate stability, high delivery efficiency, and lipophilic properties, exosomes exhibit great promise as an endogenous nano drug delivery system for delivering drugs to target tissues [105]. A number of studies have shown the significance of exosomes in the development and treatment of multiple disease [106,107,108,109,110,111,112]. We can engineer and express a single-chain-variable fragment of a high-affinity target-specific monoclonal antibody on the exosome’s surface. Subsequently, the engineered exosomes were loaded with *miR-143* and then used for the treatment of corresponding disease. Furthermore, we can encapsulate the eukaryotic expression plasmids of *miR-143* to make oral nanoparticles and applied to the treatment of some disease as described by Bao et al. [113]. Additionally, phytochemicals may provide a potential clinical therapeutic due to the ability of induce the expression of *miR-143*; for example, curcumin has been shown to significantly induce *miR-143* in cancer cells [114,115,116].

In conclusion, we are the first to report that the whole-genome methylation level could be influenced by *miR-143* knock-out in *mice* liver via WGBS, and we found that the methylation levels of many metabolic pathway-related genes were altered by *miR-143* knock out. Furthermore, *miR-143* may provide therapeutic targets for the treatment of liver damage-associated diseases and hepatocellular carcinoma, and may be a biomarker in some cancer diagnoses.

## 4. Materials and Methods

### 4.1. Sample Collection and Processing

The experiments using *mouse* materials were approved by South China Agricultural University. Global *miR-143* knock-out *mice* (FVB) were generated by Cyanogen Biosciences (Guangzhou, China) using CRISPR/CAS9 technique. A 104 bp region in the site of 5P and 3P of *Mir-143* was missing in chromosome 18 GRCm8.p6 and PCR was conducted to verify the transgenic *mice* (F 5′-TGGGTGGGTCTATCACAAGAAAGC-3′ and R 5′- GACCAGAGCTTACTGTTGTAGAGGGC -3′). Wild-type (WT) and *miR-143* knock-out (143KO) male *mice* were fed a high-fat diet (D12492) at 4 weeks of age. The diets were provided by the Guangdong Medical Laboratory Animal Center. A previous study reported that when samples were sequenced as a pool, the estimates were generally accurate, compared with separately determined samples [117]. Therefore, three individuals were mixed together for one sample in both WT and KO groups. 

Total genomic DNA was isolated and purified from frozen *mouse* liver tissue by SDS-protease K treatment, phenol extraction, and ethanol precipitation. Genomic DNA degradation and contamination was monitored on 1% agarose gels. DNA purity was checked using the NanoPhotometer^®^ spectrophotometer (IMPLEN, Calabasas, CA, USA). DNA concentration was measured using Qubit^®^ DNA Assay Kit in Qubit^®^ 2.0 Flurometer (Life Technologies, Calabasas, CA, USA). 

### 4.2. Library Preparation and Quantification

A total amount of 5.2 µg genomic DNA spiked with 26 ng lambda DNA was fragmented by sonication to 200–300 bp with Covaris S220, followed by end repair and adenylation. Cytosine-methylated barcodes were ligated to sonicated DNA as per manufacturer’s instructions. Then, these DNA fragments were treated with bisulfite twice using EZ DNA Methylation-GoldTM Kit (Zymo Research Corporation, Irvine, CA, USA), and the resulting single-strand DNA fragments were PCR amplificated using KAPA HiFi HotStart Uracil + ReadyMix (2X). Library concentration was quantified by Qubit^®^ 2.0 Flurometer (Life Technologies, Calabasas, CA, USA) and quantitative PCR, and the insert size was assayed on Agilent Bioanalyzer 2100 system.

### 4.3. Data Analysis 

After cluster generation, the library preparations were sequenced at Novogene Bioinformatics Institute (Beijing, China) on Illumina Novaseq platform and 125/150 bp paired-end reads were generated. Image analysis and base calling were performed with Illumina CASAVA pipeline, and finally 125/150 bp paired-end reads were generated.

### 4.4. Quality Control

First, we used FastQC (fastqc_v0.11.5) to perform basic statistics on the quality of the raw reads. Then, those reads produced by the Illumina pipeline in FASTQ format were pre-processed through Trimmomatic (Trimmomatic-0.36) software using the given parameters (SLIDINGWINDOW: 4:15; LEADING:3, TRAILING:3; ILLUMINACLIP: adapter.fa: 2:30:10; MINLEN:36).

The remaining reads that passed all the filtering steps were counted as clean reads on which all subsequent analyses were based. At last, we used FastQC to perform basic statistics on the quality of the clean data reads.

### 4.5. Reference Data Preparation before Analysis 

Before the analysis, we prepared the reference data for the species we study, which includes the reference sequence fasta file, the annotation file in gtf format, the GO annotation file, the description file and the gene region file in bed format. As for the bed files, we predicted repeats by RepeatMasker, followed by getting CGI track from a genome use cpgIslandExt.

### 4.6. Reads Mapping to the Reference Genome

Bismark software (version 0.16.3; [118]) was used to perform alignments of bisulfite-treated reads to a reference genome (-X 700 --dovetail). The reference genome was firstly transformed into bisulfite-converted version (C-to-T and G-to-A converted) and then indexed using bowtie2 [119]. Sequence reads were also transformed into fully bisulfite-converted versions (C-to-T and G-to-A converted) before being aligned to similarly converted versions of the genome in a directional manner. Sequence reads that produced the best and unique alignment from the two alignment processes (original top and bottom strand) were then compared to the normal genomic sequence and the methylation state of all cytosine positions in the read was inferred. The same reads that aligned to the same regions of genome were regarded as duplicated ones. The sequencing depth and coverage were summarized using deduplicated reads. 

The results of methylation extractor (bismark_methylation_extractor, --no_overlap) were transformed into bigWig format for visualization using IGV browser. The sodium bisulfite non-conversion rate was calculated as the percentage of cytosine sequenced at cytosine reference positions in the lambda genome.

### 4.7. Estimating Methylation Level 

To identify the methylation site, we modeled the sum of methylated counts (mC) as a binomial (Bin) random variable with methylation rate r
mC~Bin (mC + umC × r)

In order to calculate the methylation level of the sequence, we divided the sequence into multiple bins with the size of 10 kb. The sum of methylated and unmethylated read counts in each window was calculated. Methylation level (ML) for each window or C site shows the fraction of methylated Cs, and is defined as: $$\mathrm{ML}(C)=\frac{\mathrm{reads}(\mathrm{mC})}{\mathrm{reads}(\mathrm{mC})+\mathrm{reads}(C)}$$

Calculated ML was further corrected with the bisulfite non-conversion rate according to previous studies [120]. Given a bisulfite non-conversion rate r, the corrected ML was estimated as: $${\mathrm{ML}}_{\left(\mathrm{corrected}\right)}=\frac{\mathrm{ML}-r}{1-r}$$

### 4.8. Differentially Methylated Analysis 

Differentially methylated regions (DMRs) were identified using the DSS software [121,122,123]. The core of DSS is a new dispersion shrinkage method for estimating the dispersion parameter from Gamma-Poisson or Beta-Binomial distributions. 

DSS possesses three approaches to detect DMRs. The first is spatial correlation. Proper utilization of the information from neighboring cytosine sites can help improve estimation of methylation levels at each cytosine site, and hence improve DMR detection. Second, the read depth of the cytosine sites provides information on precision that can be exploited to improve statistical tests for DMR detection. Finally, the variance among biological replicates provides information necessary for a valid statistical test to detect DMRs. When there is no biological replicate, DSS combines data from nearby cytosine sites and uses them as ‘pseudo-replicates’ to estimate biological variance at specific locations. DMRs identification parameters: smoothing = TRUE, smoothing.span = 200, delta = 0, p.threshold = 0.00001, minlen = 50, minCG = 3, dis.merge = 100, pct.sig = 0.5. 

According to the distribution of DMRs in the genome, we defined the genes related to DMRs as genes whose gene body region (from TSS to TES) or promoter region (upstream 2kb from the TSS) had an overlap with the DMRs. 

### 4.9. GO and KEGG Enrichment Analysis of DMR-Related Genes 

Gene Ontology (GO) enrichment analysis of genes related to DMRs was implemented by the GOseq R package [63], in which gene length bias was corrected. GO terms with corrected *p*-value less than 0.05 were considered significantly enriched by DMR-related genes. 

KEGG [65] is a database resource for understanding high level functions and utilities of the biological system, such as the cell, the organism and the ecosystem, from molecular-level information, especially large-scale molecular datasets generated by genome sequencing and other high- through put experimental technologies (http://www.genome.jp/kegg/, accessed date: 7 April 2019). We used KOBAS software [124] to test the statistical enrichment of DMR related genes in KEGG pathways.

### 4.10. Gene Expression Analysis by Quantitative RT-PCR

Total RNAs were extracted from the testis and ovary tissue using TRIzol reagent (Invitrogen, Carlsbad, CA, USA) according to the manufacturer’s instructions. After treatment with DNase I (Takara Bio Inc., Shiga, Japan), total RNA (1.5 μg) was reverse-transcribed to cDNA in a final 20 μL using M-MLV Reverse Transcriptase (Promega, Madison, WI, USA) plus RNase inhibitor (Promega, Shanghai, China) with oligo-d(T)s or loop as primers. SYBR Green Real-time q-PCR Master Mix reagents (Promega, Madison, WI, USA), sense and antisense primers were used for real-time quantitative polymerase chain reaction (RT-qPCR) analysis, which was performed using CFX96 Touch™ Optics Module instrument (BIO-RAD, California, CA, USA). *U6* was used as a candidate housekeeping gene. The following primers were designed: *U6*, F 5′-CTCGCTTCGGCAGCACA -3′ and R 5′-AACGCTTCACGAATTTGCGT-3′; *miR-143*, loop Primer 5′-GTCGTATCCAGTGCGTGTCGTGGAGTCGGCAATTGCACTGGATACGACGAGCTA-3′, F 5′-GGGTGAGATGAAGCACTG-3′ and R 5′-CAGTGCGTGTCGTGGAGT-3′.

### 4.11. Statistical Analysis

All data were expressed as the mean ± standard error of the mean (SEM). Statistical differences among groups were obtained using T tests (SPSS 22.0, Chicago, IL, USA). *p* < 0.05 was considered statistically significant.

## Figures and Tables

**Figure 1 ijms-22-13075-f001:**
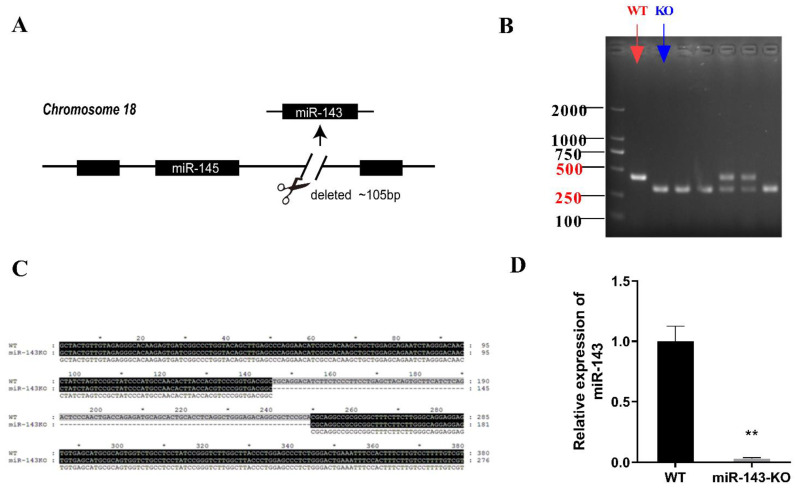
Identification of transgenic *mice*. (**A**) The schematic diagram of transgenic *mice*. (**B**) The electrophoretic bands of *miR-143* KO *mice* and WT *mice* after gDNA PCR. (**C**) Gene sequence alignment of *miR-143* KO *mice* and WT *mice*. (**D**) The relative expression levels of *miR-143* in the liver of WT and *miR-143* KO *mice*. (* *p* < 0.05, ** *p* < 0.01, n = 8).

**Figure 2 ijms-22-13075-f002:**
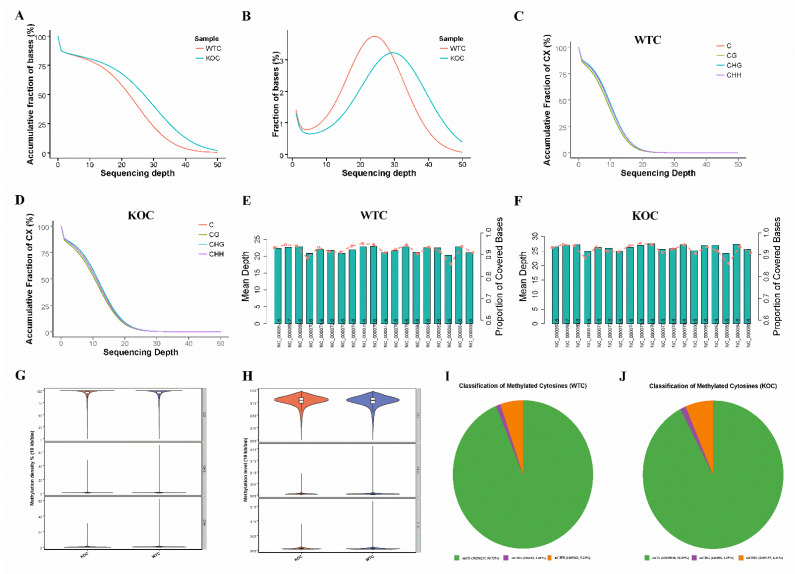
Whole-Genome Bisulfite Sequencing of liver sample in WT and KO *mice*. (**A**) The cumulative genome distribution in the liver of WT and KO *mice*. (**B**) The genome coverage of the liver in WT and KO *mice*. (**C**,**D**) The cumulative coverage of C sites in each context. (**E**,**F**) Distribution of WGBS reads on chromosomes. (**G**) Whole-genome DNA methylation density in each context. (**H**) The mean DNA methylation level in each context. (**I**,**J**) The ratio of methylated cytosines in each context (green: mCG, purple: mCHG, and orange: mCHH).

**Figure 3 ijms-22-13075-f003:**
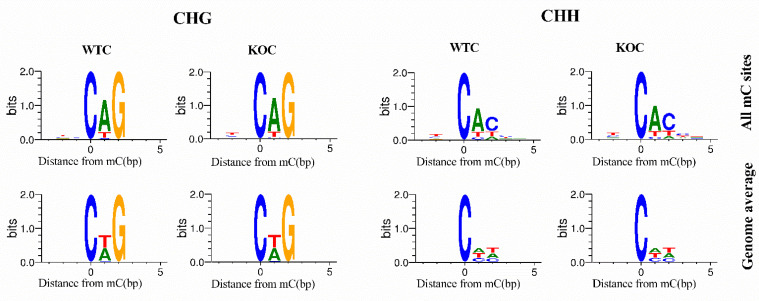
Logo plots of the sequences proximal to sites of non-CG DNA methylation in each sequence context. The abscissa axis indicates base positions and the ordinate axis indicates the degree of base enrichment. Four different colors represent the different bases (color version of this figure is available at Bioinformatics online).

**Figure 4 ijms-22-13075-f004:**
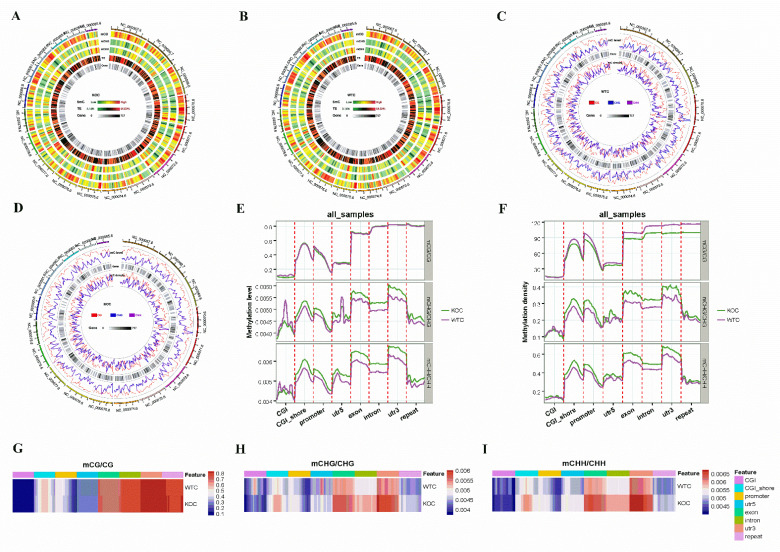
Genome-wide DNA methylation profiling in liver samples. (**A**,**B**) Density plot of 5-methylcytosine in sequence contexts (mCG, mCHG, mCHH where mC signifies 5-methylcytosine; H = A, C or T), transposable elements and genes. (**C**,**D**) The distribution of gene density and DNA methylation level across chromosomes. (**E**,**F**) The methylation level (**E**) and density (**F**) of cytosine in different contexts (CG, CHG, CHH) in featured regions of the genome. (**G**–**I**) The heat map of the methylation levels of mCG (**G**), mCHG (**H**), and mCHH (**I**) in the gene functional regions.

**Figure 5 ijms-22-13075-f005:**
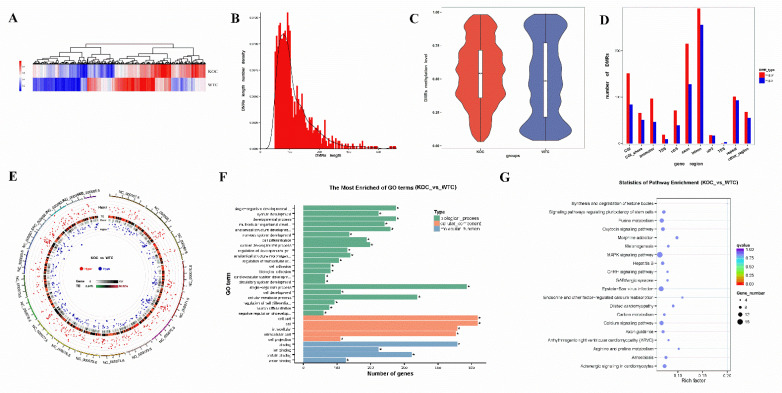
The characters of DMRs. (**A**) DMR cluster analysis using heat map. (**B**) Distribution of DMR lengths. (**C**) The methylation level of DMRs in KO and WT groups. (**D**) The number of DMR distributed in CGI, CGI shore, promoter, TSS, 5′ UTRs, exons, introns, 3′UTRs, TES, repeats and other regions. (**E**) Circos plots of the distribution and statistical significance of DMRs in each chromosome. The red and blue dots indicated hypermethylated and hypomethylated DMRs, and one larger in size and outer in position indicated a more significant difference. (**F**) Most enriched GO terms. (**G**) The results of KEGG pathway enrichment.

**Figure 6 ijms-22-13075-f006:**
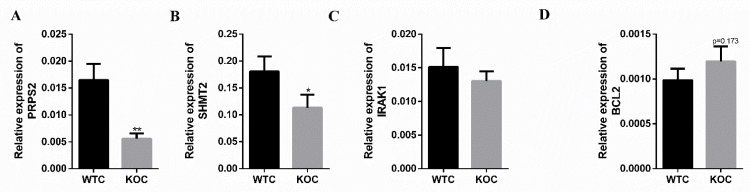
The expression of DMR-related genes. (**A**–**D**) Relative mRNA levels of *Prps2* (**A**), *Shmt2* (**B**), *Irak1* (**C**), *Bcl2* and (**D**) (* *p* < 0.05, ** *p* < 0.01, *n* = 8).

**Table 1 ijms-22-13075-t001:** The overview of output data by Whole-Genome Bisulfite Sequencing.

Samples	WTC	KOC
Raw Reads	389,070,805	418,284,236
Raw Bases(G)	116.72	125.49
Clean Reads	380,970,003	408,819,442
Clean Bases(G)	104.02	111.29
Clean_ratio (%)	89.12	88.68
Q20(%)	96.36	96.32
Q30(%)	89.63	89.48
GC Content (%)	21.46	21.69
BS Conversion Rate (%)	99.802	99.807
Mapped Reads	259,554,863	296,026,157
Unique Mapping Rate (%)	68.13	72.41
Duplication Rate (%)	19.17	15.13
Number of Sites	2,467,496,725	2,469,014,115
1× Coverage (%)	87.53	87.59
5× Coverage (%)	83.58	84.04
10× Coverage (%)	79.02	80.62
C(Mb)	10,010.9	11,945.9
CG(Mb)	383.3	462.4
CHG(Mb)	2150.3	2594.3
CHH(Mb)	7477.2	8889.3
MeanC (%)	3.32	3.39
MeanCG (%)	74.71	74.99
MeanCHG (%)	0.45	0.47
MeanCHH (%)	0.49	0.52

## Data Availability

All sequencing raw data sets were deposited into the National Center for Biotechnology Information (NCBI) Sequence Read Archive (SRA) database under BioProject accession number PRJNA71957 (https://dataview.ncbi.nlm.nih.gov/object/PRJNA719576). Sequencing files can be found under accession number SRR14140298 and SRR14140299.

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
