# Peer review of "Exploration of the Effect on Genome-Wide DNA Methylation by miR-143 Knock-Out in Mice Liver"

_ijms, 2021, doi:10.3390/ijms222313075_

Round 1

Reviewer 1 Report

It's a nice paper. It thus should be published, pending native English proofing

Author Response

Dear Editors and Reviewers:

Thank you very much for sending us the Reviewers’ reports on our manuscript (ijms-1458839) entitled “Exploration of the effect on genome-wide DNA methylation by miR-143 knock-out in mice liver”. Particularly, we would like to thank the Reviewers for their valuable comments and criticisms.

According to the Reviewers and Editors’ recommendations, we have revised carefully our manuscript with "track changes". We hope the revised manuscript will meet your journal’s standard. Below you will find our point-by-point responses to the reviewers’ comments/questions and changes the authors have made. If any question arises, please let us know.

Comments: It's a nice paper. It thus should be published, pending native English proofing.

Response: Yes, that’s a very good advice. According to your advice, we have proceeded English editing, please see in revised manuscript. It is greatly appreciated for your advice! Thank you very much!

Thank you very much for your consideration.

Yours faithfully,

Qian-yun, Xi

College of Animal Science, South China Agriculture University, 483 Wushan Road, Guangzhou 510642, China.

Reviewer 2 Report

In this manuscript by Chen etal the authors  have tried  to dissect the differentially methylated regions, and correlating between genomic DNA methylation and gene expression of the genome, deciphering their relationship to miR-143 versus methylation. Ras genes are important targets of miR-143 but does not appear discussion on that topic. The authors should provide significant discussion on that topic. Authors have proposed possible use of miR-143 to correct some of the diseases that result due to loss of miR-143 as cause of the disease. It will be pertinent to add some discussion referirng to reports wherein they have used modified miR-143 as stable forms of the miRNAs for treatment purposes. Additionally, phytochemicals such as curcumin have also been shown to significantly induce miR-143 in cancer cells. The manuscript needs careful editing throughout the manuscript before it may be considered of   publication quality

Author Response

Dear Editors and Reviewers:

Thank you very much for sending us the Reviewers’ reports on our manuscript (ijms-1458839) entitled “Exploration of the effect on genome-wide DNA methylation by miR-143 knock-out in mice liver”. Particularly, we would like to thank the Reviewers for their valuable comments and criticisms.

According to the Reviewers and Editors’ recommendations, we have revised carefully our manuscript with "track changes". We hope the revised manuscript will meet your journal’s standard. Below you will find our point-by-point responses to the reviewers’ comments/questions and changes the authors have made. If any question arises, please let us know.

Our point-by-point responses to your comments/questions please see the attachment.

Reviewer 3 Report

The authors proposed an interesting manuscript aimed at evaluating the involvement of miR-143 in the regulation of DNA methylation. For this purpose, the authors knocked down the coding sequence of miR-143 and analyze the global methylation status of liver cells obtained from mice WT and KO for miR-143. Through both experimental and computational approaches, the authors demonstrated that the KO of this miRNA is responsible for the alteration of the methylation status of different genes involved in the alteration of key cellular and molecular functions. Overall, the study is very interesting and gives novel information on the relationship existing between two different epigenetic phenomena. However, there are some issues that the authors have to address before publication:
1) The manuscript needs English editing before publication;
2) In the abstract section it is not clear what mean WT group and KO groups. Which gene or factor was knocked down?
3) It is not clear why the authors did not evaluate the expression levels of the direct target of miR-143. In particular, the authors did not perform any computational or experimental analysis on the mRNA targets of miR-143. For example, it would be interesting to evaluate the expression levels of DNMT3 and other DNMT family members in both WT and KO samples in order to evaluate if the KO of miR-143 directly influences DNA methylation through the alteration of DNMT genes. Please address this fundamental issue;
4) In the Discussion section the authors state: “However, there are few reports about the relationship between miRNA and the methylation level of the genome.”. Recently some studies have investigated the epigenetic alterations induced by chemicals and toxicants which mediate the alteration of the expression levels of miRNAs and the DNA methylation trying to integrate these different epigenetic modifications and associate them with the development of diseases. Please argue this aspect or provide references supporting this notion. For this purpose, see:
- 34444445
- 33926515
- 24976726
5) Remove the following paragraph: “Conclusions This section is not mandatory but can be added to the manuscript if the discussion is unusually long or complex.”.

Author Response

(The authors gave the same response as above.)

Reviewer 4 Report

Dear, Authors

The work showed in this article by the authors is a very high-level study of miR143 knockout mice fed with HFD, analyzed using bioinformatics research methods. The authors proved that the loss of miR143 increases the methylation of cytosine in the whole genome of mouse liver by 0.19% and also analyzed the changes in detail and showing very valuable and voluminous data.

My comments and suggestions can be found in the attached PDF. I would be happy to contribute to the development of your future work.

Sincerely yours,

Author Response

Dear Editors and Reviewers:

Thank you very much for sending us the Reviewers’ reports on our manuscript (ijms-1458839) entitled “Exploration of the effect on genome-wide DNA methylation by miR-143 knock-out in mice liver”. Particularly, we would like to thank the Reviewers for their valuable comments and criticisms.

According to the Reviewers and Editors’ recommendations, we have revised carefully our manuscript with "track changes". We hope the revised manuscript will meet your journal’s standard. Below you will find our point-by-point responses to the reviewers’ comments/questions and changes the authors have made. If any question arises, please let us know.

Our point-by-point responses to your comments/questions please see the attachment.

Thank you very much for your consideration.

Yours faithfully,

Qian-yun, Xi

College of Animal Science, South China Agriculture University, 483 Wushan Road, Guangzhou 510642, China.

E-mail Address: [email protected]

Round 2

Reviewer 3 Report

The authors have significantly improved their manuscript. They well-addressed all of my comments. The paper can be accepted for publication after the editorial check.

Author Response

Dear Editors and Reviewers:

Thank you very much for sending us the Reviewers’ reports on our manuscript (ijms-1458839) entitled “Exploration of the effect on genome-wide DNA methylation by miR-143 knock-out in mice liver”. Particularly, we would like to thank the Reviewers for their valuable comments and criticisms.

According to the Reviewers and Editors’ recommendations, we have revised carefully our manuscript with "track changes". We hope the revised manuscript will meet your journal’s standard. Below you will find our point-by-point responses to the reviewers’ comments/questions and changes the authors have made. If any question arises, please let us know.

Comments:The authors have significantly improved their manuscript. They well-addressed all of my comments. The paper can be accepted for publication after the editorial check.

Response: Thank you for your questions and suggestions. It makes our paper better. It is greatly appreciated for your questions and suggestions! Thank you very much!

Thank you very much for your consideration.

Yours faithfully,

Qian-yun, Xi

College of Animal Science, South China Agriculture University, 483 Wushan Road, Guangzhou 510642, China.

E-mail Address: [email protected]

Reviewer 4 Report

Dear authors,

The authors have changed partially the previous manuscript to reflect the reviewer's suggestions and comments. The revised version of the manuscript is gotten better than before.

My comments and suggestions can be found in the attached PDF. I would be happy to contribute the development of your future work

Sincerely yours,

Author Response

Dear Editors and Reviewers:

Thank you very much for sending us the Reviewers’ reports on our manuscript (ijms-1458839) entitled “Exploration of the effect on genome-wide DNA methylation by miR-143 knock-out in mice liver”. Particularly, we would like to thank the Reviewers for their valuable comments and criticisms.

According to the Reviewers and Editors’ recommendations, we have revised carefully our manuscript with "track changes". We hope the revised manuscript will meet your journal’s standard. Below you will find our point-by-point responses to the reviewers’ comments/questions and changes the authors have made. If any question arises, please let us know.

A point-by-point response to your comments and suggestions, please see the attachment.

 Thank you very much for your consideration.

Yours faithfully,

Qian-yun, Xi

College of Animal Science, South China Agriculture University, 483 Wushan Road, Guangzhou 510642, China.

Round 3

Reviewer 4 Report

Dear authors,

The authors have revised the manuscript.

However, there is no mention of uric acid metabolism and XDH in the intro, and the appearance of those results is too abrupt and completely incomprehensible.

It does not seem to make much sense to add these statements in the intro, so please delete the statements about them.

Specifically, please delete all the content related to results 2.7 and Fig6E-H in the results.

I hope the progress of the authors' work.

Sincerely yours,

Author Response

Dear Editors and Reviewers:

Thank you very much for sending us the Reviewers’ reports on our manuscript (ijms-1458839) entitled “Exploration of the effect on genome-wide DNA methylation by miR-143 knock-out in mice liver”. Particularly, we would like to thank the Reviewers for their valuable comments and criticisms.

According to the Reviewers and Editors’ recommendations, we have revised carefully our manuscript with "track changes". We hope the revised manuscript will meet your journal’s standard. Below you will find our point-by-point responses to the reviewers’ comments/questions and changes the authors have made. If any question arises, please let us know.

 Comments:The authors have revised the manuscript. However, there is no mention of uric acid metabolism and XDH in the intro, and the appearance of those results is too abrupt and completely incomprehensible. It does not seem to make much sense to add these statements in the intro, so please delete the statements about them. Specifically, please delete all the content related to results 2.7 and Fig6E-H in the results. I hope the progress of the authors' work.

Response: Yes, we agree with your suggestion. According to your advice, we have deleted all the content related to results 2.7 and Fig6E-H in the results (Lines 233-252 and Lines 286-294). In addition, we also deleted the discussion and methods of the results, such as the keywords “Gout” (Lines 39), the discussion (Lines 367-370, Lines 376-379 and Lines 414), materials and methods (Lines 524-527 and Lines 542-574), please see Lines 39, Lines 233-252, Lines 286-294, Lines 367-370, Lines 376-379, Lines 414, Lines 524-527 and Lines 542-574 in revised manuscript. It is greatly appreciated for your advice and wishes! Thank you very much!

Thank you very much for your consideration.

Yours faithfully,

Qian-yun, Xi

College of Animal Science, South China Agriculture University, 483 Wushan Road, Guangzhou 510642, China.

E-mail Address: [email protected]